# Microstructures and Fatigue Properties of High-Strength Low-Alloy Steel Prepared through Submerged-Arc Additive Manufacturing

**DOI:** 10.3390/ma15238610

**Published:** 2022-12-02

**Authors:** Mei-Juan Hu, Ling-Kang Ji, Qiang Chi, Qiu-Rong Ma

**Affiliations:** CNPC Tubular Goods Research Institute, State Key Laboratory for Performance and Structure Safety of Petroleum Tubular Goods and Equipment Materials, Xi’an 710077, China

**Keywords:** submerged-arc additive manufacturing, high-strength low-alloy steel, low-cycle fatigue, fatigue crack propagation rate

## Abstract

Submerged arc additive manufacturing (SAAM) is a viable technique for manufacturing large and complex specialized parts used in structural applications. At present, manufacturing high-strength low-alloy steel T-branch pipe through SAAM has not been reported. This paper uses this technology to manufacture low-alloy structural steel parts. The microstructures of the samples were characterized, which revealed that they were mainly composed of polygonal ferrites. The tensile properties in the horizontal and vertical directions of deposits were studied. Results show that the horizontal tensile strength of deposits was quite close to the vertical one, while the elongation rate in the vertical direction was obviously lower than that in the horizontal direction. Fatigue results indicate that the strain fatigue limit of high-strength low-alloy steel samples in vertical direction was 0.24%. The fatigue fractures of fatigue samples of deposits showed multi-source crack initiation characteristics and the crack propagation regions exhibited typical fatigue striations, so the final instantaneous fracture region showed a ductile fracture. Fatigue performance is very important for the safe service of structural parts, but there is a lack of relevant research on this additive manufacturing part. The results of this paper may support the popularization of the SAAM for high-strength low-alloy steel T-branch pipe.

## 1. Introduction

Pipelines made of high-strength low-alloy steel are the safest and most economic mode to transport oil and gas. In the last thirty years, the development and usage of high-grade pipeline steel have been a hot topic in pipeline construction to meet the energy demand of society and industrial production, and further improve the transportation efficiency of pipelines [1,2]. Scholars have developed high-grade pipeline steel including X70 and X80 in succession and used them for natural gas transportation. In addition, high-grade pipeline steel X100 and X120 have been developed and tried for application. However, considering both the economy and safety of pipeline transportation, X80 pipeline steel is still the mainstream steel grade for producing pipelines for natural gas transportation in the world [3,4].

With the increasingly wide application of X80 pipeline steel in pipeline transportation, the manufacturing of steel has become an extremely crucial link [5,6]. Mohd NA et al., made a very systematic summary of the advantages and disadvantages of additive manufacturing technology [7]. Different from traditional manufacturing modes, wire and arc additive manufacturing (WAAM) developed relatively rapidly in recent years, providing a new idea for manufacturing X80 pipeline steel. WAAM is an additive, manufacturing technology through layer-by-layer stacking, taking the welding arc as the heat source and wires as raw materials. The technology is mainly advantageous, having a high deposition rate, a low manufacturing cost, and a high degree of freedom of machining, so it is particularly suitable for additive manufacturing in fields including aerospace and pressure vessels [8,9].

Many scholars in China and abroad have studied the microstructures and mechanical properties of WAAM workpieces. They find that the traditional WAAM components are faced with a problem pertaining to the inhomogeneity of microstructures and mechanical properties in various directions [10,11]. Therefore, the anisotropy of workpieces needs to be eliminated through techniques such as subsequent thermal processing, so as to improve mechanical properties, which therefore greatly reduces the production efficiency and increases the manufacturing cost when producing thick-walled components [12,13].

To solve the problem, Chen et al., carried out submerged-arc additive manufacturing (SAAM) taking submerged-arc welding with high cladding efficiency as the heat source, thus producing low-carbon deposited walls. Results show that due to the unique in-situ intrinsic heat treatment (IHT) typical of SAAM, that is, combining multi-layer penetration normalizing, full-layer-penetration inter-critical annealing, and long-duration tempering, the middle part of deposited walls is mainly composed of completely equiaxed ferrites that are homogeneous in the vertical direction. The Charpy impact toughness of the middle part at −60 °C may exceed 300 J and be isotropic [14]. Li et al., studied the microstructures and mechanical properties in various directions of high-strength Mn-Ni-Mo steel for producing components of nuclear power plants. Compared with WAAM components, the mechanical properties of SAAM components exhibit lower anisotropy and better strength-toughness balance. Therefore, the comprehensive performance of SAAM components is superior to steel components of nuclear power plants and marine engineering manufactured using some traditional methods. Their analysis indicates that steel components manufactured through SAAM are strengthened mainly via grain-boundary strengthening and solid-solution strengthening. Compared with as-forged components at the same grade, the high impact toughness and the low tough-brittle transition temperature are attributed to smaller sizes of grains and martensite/austenite islands. Their research results indicate that SAAM not only can machine large and medium components but also confers relatively uniform microstructures and mechanical properties to additively manufactured steel samples [15].

Natural gas pipelines in service are under alternating stress induced by internal pressure fluctuations and external load changes. As the service time increases, fatigue cracks initiate and even fatigue fractures occur in pipelines under alternating stress [16]. Among fracture modes of engineering structures, fatigue fractures account for a large proportion, particularly in natural gas pipelines under alternating stress, for which fatigue fractures account for more than 60% of total fracture modes. Moreover, no large plastic deformation is observed from initiation to propagation of fatigue cracks, so it is challenging to detect fatigue fractures. Therefore, fatigue fracture-induced accidents generally cause larger economic losses. How to prevent the occurrence of fatigue cracks and improving the fatigue life of pipelines have become key problems for the safety of pipelines [17].

At present, manufacturing high-strength low-alloy steel T-branch pipe through SAAM has not been reported. In the research, high-strength low-alloy steel T-branch pipe with similar chemical composition to that of X80 pipeline steel was prepared via SAAM, and the microstructures and fatigue properties of the SAAM components were studied. Research results may support the popularization of the SAAM for high-strength low-alloy steel T-branch pipe.

## 2. Materials and Methods

### 2.1. Material Compositions and Process Parameters

The chemical compositions of welding wires used for preparing high-strength low-alloy steel T-branch pipe through SAAM are listed in Table 1. The additive manufacturing parameters of high-strength low-alloy steel T-branch pipe via SAAM are shown in Table 2. The HSLA grade is 550 MPa and the diameter of the welding wire is 4 mm.

### 2.2. Microstructures in Deposits of SAAM Workpieces

The observation site of microstructures is displayed in Figure 1. In the test, 4% nitric acid alcohol was used as the etchant and the corrosion lasted for 5 s. The test equipment was a Nikon ECLIPSE MA200 inverted metallographic microscope. In high-resolution scanning electron microscope SEM tests, the etchant was the same as was used for the metallographic samples, while the corrosion time was 10 s, which was 5 s longer than that of the metallographic samples. The test equipment was an FEI Verios460 high-resolution SEM.

### 2.3. Tensile Tests on Deposits of SAAM Workpieces

The tensile properties of SAAM workpieces in two directions were explored, and the sampling directions are shown in Figure 1. To ensure the accuracy of data, three samples were taken in each direction and then tested. The tensile tests were conducted following the national standard *Method of tensile test for welded joint* (GB/T 2651–2008). Sample dimensions are illustrated in Figure 2. The tensile tests were performed on an INSTRON 1195 electronic tensile tester at a rate of extension of 0.5 mm/min.

### 2.4. Low-Cycle Fatigue Tests on Deposits of SAAM Workpieces

The blank samples were cut along the vertical direction of SAAM workpieces at the positions shown in Figure 1. Sample dimensions are illustrated in Figure 3. The strain amplitudes are separately 0.8%, 0.6%, 0.5%, 0.4%, 0.3%, and 0.25%. In this experiment, when the strain amplitude was 0.6%, 0.5%, and 0.4%, two samples were tested. When the strain amplitudes were 0.8%, 0.3%, and 0.25%, a sample was tested. The tests were carried out following the national standard The test method for axial loading constant-amplitude low-cycle fatigue of metallic materials (GB/T15248–2008). The axial loading with the axial strain controlled was adopted, and the strain ratio was R = −1. The triangular waves were applied and the strain rate was 5 × 10^−3^ S^−1^. The frequency in the experiment was 0.5 Hz per second.

The tests were conducted in the atmospheric environment at room temperature. Fatigue fractures at three strain levels were selected for observation. A SU3500 tungsten filament SEM was used to observe fracture morphologies of low-cycle fatigue samples.

### 2.5. Tests on the Fatigue Crack Propagation Rate of Deposits of SAAM Workpieces

The sampling position for tests on the fatigue crack propagation rate in SAAM workpieces is shown in Figure 1. The tests were performed following the *Standard test method for fatigue crack growth rates of metallic materials* (GB/T6398-2000) on an INSTRON-1341 testing machine at room temperature in the atmospheric environment. The loading mode with axial stress control was adopted, and the loading frequency, stress ratio, and waveform were 20 Hz, R = 0.1, and sine waves, respectively. Figure 4 illustrates the sample dimensions for testing the fatigue crack propagation rate of components.

## 3. Test Results and Analysis

### 3.1. Microstructures in Deposits of SAAM High-Strength Low-Alloy Steel Workpieces

Figure 5 shows the morphologies of microstructures on the vertical section of deposits of the SAAM workpieces. Figure 5a displays the macroscopic morphology of samples and the sampling position is shown in Figure 1. Figure 5b illustrates the metallographic microstructures in deposits of the SAAM high-strength low-alloy steel workpieces. The bright white area is found to have a large number of polygonal ferrites, which are generally formed at a high transformation temperature and a slow cooling rate. Previous research has shown that the structure nucleates at the original austenite grain boundary in priority and its growth can cross such a boundary, thus covering the outline of original austenite grain boundaries [18]. Figure 5c shows the morphology of microstructures in deposits of SAAM high-strength low-alloy steel workpieces under the SEM. In the figure, polygonal ferrites are dark gray, the white grain boundaries seem like interlaced networks, and polygonal ferrites with the size of 6~15 μm have regular and smooth boundaries. Figure 5d is a locally enlarged picture of Figure 5, in which white blocky pearlites can be found in local areas.

### 3.2. Tensile Properties of Deposits of SAAM High-Strength Low-Alloy Steel Workpieces

The horizontal (X) and vertical (Y) tensile curves of the high-strength low-alloy steel T-branch pipe prepared through SAAM are illustrated in Figure 6. The test results show that the tensile strengths of the deposits in the horizontal (X) and vertical (Y) directions are separately 705 and 695 MPa. The elongation rates of samples in the two directions are calculated and the elongations at the break of the deposits in the horizontal (X) and vertical (Y) directions are 27% and 22%, respectively. The result indicates that the tensile strengths differ slightly in the two directions, while the plasticity of the deposits in the vertical direction is weaker than that in the horizontal direction.

### 3.3. Low-Cycle Fatigue Properties of Deposits of SAAM High-Strength Low-Alloy Steel Workpieces

Figure 7 illustrates the relationship curve between the strain amplitude and the cycle life of SAAM high-strength low-alloy steel workpieces. The cycle life of SAAM workpieces under strain amplitudes of 0.8%, 0.6%, 0.5%, 0.4%, 0.3%, and 0.25% was investigated. Table 3 lists the fatigue test data of the SAAM workpieces under different strain amplitudes.

The Zheng-Hirt formula is used to fit the data of low-cycle fatigue tests of SAAM high-strength low-alloy steel workpieces and the fatigue curves are shown in Figure 7. In this way, the fitting equation with the strain fatigue limit is obtained, as expressed below:(1)$$N_{f}=27.9\times {\left(\Delta \varepsilon_{t}-0.0.24\right)}^{-2}$$
where $N_{f}$ and $\Delta \varepsilon_{t}$ separately represent the number of cycles and the strain amplitude. Equation (1) is the expression for the strain amplitude of the base metal of high-strength steel and the number of cycles, with a correlation coefficient of 0.91. It can be seen from Equation (1) that the strain fatigue limit of SAAM high-strength low-alloy steel workpieces is 0.24%.

Changes in the amplitude of tensile stress of SAAM high-strength low-alloy steel workpieces under different strain amplitudes are illustrated in Figure 8. The samples all show fatigue characteristics of cyclic softening in the low-cycle fatigue tests. At first, the amplitude of tensile stress corresponding to the initial life cycles (10% of the total life cycles) may reduce under all strain amplitudes, except for the maximum strain amplitude of 0.8%. Then, the amplitude of tensile stress decreases slowly in 80% of total life cycles after reaching a stable number of cycles. Finally, the amplitude of tensile stress decreases abruptly in the remaining 10% of total life cycles until the failure of samples [19].

Figure 9 illustrates the morphologies of low-cycle fatigue fractures of SAAM high-strength low-alloy steel workpieces under different strain amplitudes. Figure 9a–d separately show the macroscopic morphologies of fatigue fractures under strain amplitudes of 0.3%, 0.5%, 0.6%, and 0.8%, and no defects such as macropores and inclusions are found in macroscopic morphologies of joints. The fractures are mainly divided into two characteristic regions: the upper part is the fatigue region, which is smooth, and the remaining is the final tensile fracture region. Figure 9e–h display the initiation positions of fatigue cracks in samples under various strain amplitudes, in which circles mark the initiation sources of fatigue cracks. Fatigue cracks in the two circles in each figure both initiate from the sample surfaces. The SAAM high-strength low-alloy steel workpieces show multi-source crack initiation characteristics in the process of low-cycle fatigue tests [20]. Figure 9i–l display the fatigue crack propagation regions in the SAAM high-strength low-alloy steel workpieces in the process of low-cycle fatigue tests. Obvious fatigue striations are found in these regions under each strain amplitude. Cracks are opened under cyclic tensile stress, while cracks destabilize under compressive stress in reverse loading, and grooves are formed at crack tips. Finally, new cracks are formed under the maximum cyclic compressive stress, while their length has increased [21]. After repeating in this way, cracks constantly propagate forward. Figure 9m–p shows the final tensile fracture regions of SAAM high-strength low-alloy steel workpieces, in which lots of dimples are observed, indicative of favorable ductility.

### 3.4. The Fatigue Crack Propagation Rate in Deposits of Saam High-Strength Low-Alloy Steel Workpieces

The relationship curve between the vertical fatigue crack propagation rate $\frac{da}{dN}$ of deposits of SAAM workpieces and the stress intensity factor (SIF) amplitude ΔK is illustrated in Figure 10. The fatigue crack propagation rate and the SIF amplitude ΔK show a double-log linear relationship on the whole. As the SIF amplitude constantly grows, the vertical fatigue crack propagation rate of deposits continues to increase.

The Paris equation is adopted to fit test data of the vertical fatigue crack propagation rate of deposits, as expressed below [22]:(2)$$\frac{da}{dN}=1.05\times 10^{-7}(\Delta {K)}^{2.24}$$

Some studies consider the exponent in the Paris equation as the Paris exponent. As the SIF amplitude constantly rises, fatigue cracks in materials propagate fast under conditions of a large Paris exponent, that is, the materials show weak resistance to fatigue crack propagation [23].

Figure 11 shows fracture morphologies of samples for testing the fatigue crack propagation rate of deposits of SAAM workpieces. Figure 11a illustrates the overall macroscopic morphology of the sample, which is divided into four regions: the initiation region, stable propagation region, and rapid propagation region of fatigue cracks, as well as the final tensile fracture region. Figure 11b displays the initiation region of fatigue cracks, which shows the multi-source crack initiation characteristics. Figure 11c illustrates the stable propagation region of fatigue cracks, in which the fractures are smooth due to the alternate loading and unloading of tensile stress, so it is a typical fatigue region. Figure 11d shows the rapid propagation region of fatigue cracks, which is the transition region between fatigue and fracture regions, in which the tunneling effect occurs at the crack tips, as indicated by red arrows in Figure 11d. The red arrow points to the leading edge of the crack. Some scholars consider that the middle position (approximately plane strain state) of through-going cracks under the fatigue load is generally ahead of the surface (approximately plane stress state), thus forming bent crack tips. The phenomenon is termed as the tunneling effect [24]. Figure 11e shows the final tensile fracture region, in which the fracture surfaces are relatively rough.

## 4. Conclusions

How to control the interlayer temperature of multi-layer multi-channel submerged arc welding additive and obtain a relatively uniform microstructure in all directions is one of the key factors to improve the fatigue performance of additive manufacturing parts.

The low-cycle fatigue properties and the fatigue crack propagation rate of deposits of SAAM high-strength low-alloy steel workpieces were studied and the microstructures in the deposits were observed. The tensile properties of the deposits in two directions were compared and the morphologies of fatigue fractures of the deposits were analyzed. Finally, the following conclusions are obtained:
(1)The deposits of SAAM high-strength low-alloy steel workpieces are mainly composed of polygonal ferrites, which have regular and smooth grain boundaries and sizes of 6~15 μm.(2)The horizontal and vertical tensile strengths of the deposits of SAAM high-strength low-alloy steel workpieces are 705 and 695 MPa, respectively. The plasticity of the deposits in the vertical direction is weaker than that in the horizontal direction.(3)The strain fatigue limit of the deposits of SAAM high-strength low-alloy steel workpieces is 0.24%, and the deposits show cyclic softening characteristics in the fatigue tests.(4)Observation of the morphologies of fatigue fractures in deposits of SAAM high-strength low-alloy steel workpieces reveals that each region of the joints shows multi-source crack initiation characteristics. Fatigue cracks are found in all propagation regions of fatigue cracks. Lots of dimples are observed in the final tensile fracture region, indicative of ductile fracture characteristics.

## Figures and Tables

**Figure 1 materials-15-08610-f001:**
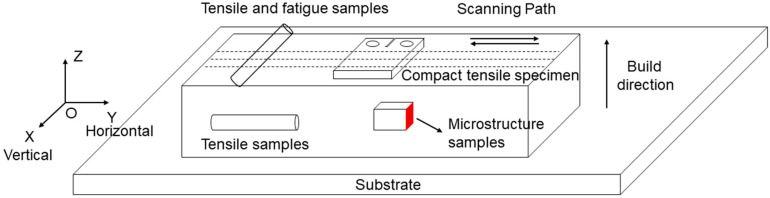
Sampling of multi-bead multi-layer blocks.

**Figure 2 materials-15-08610-f002:**
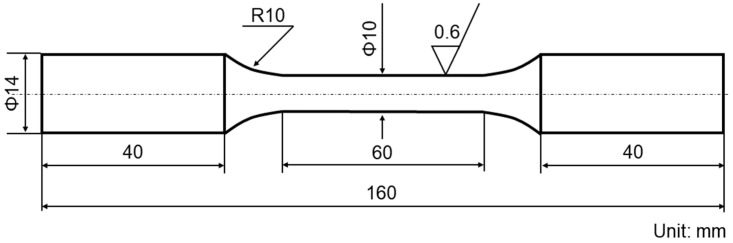
Dimensions of tensile samples.

**Figure 3 materials-15-08610-f003:**
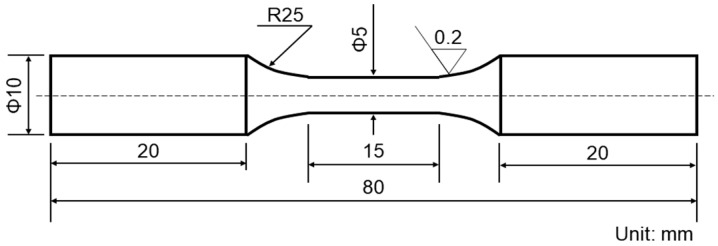
Dimensions of low-cycle fatigue samples.

**Figure 4 materials-15-08610-f004:**
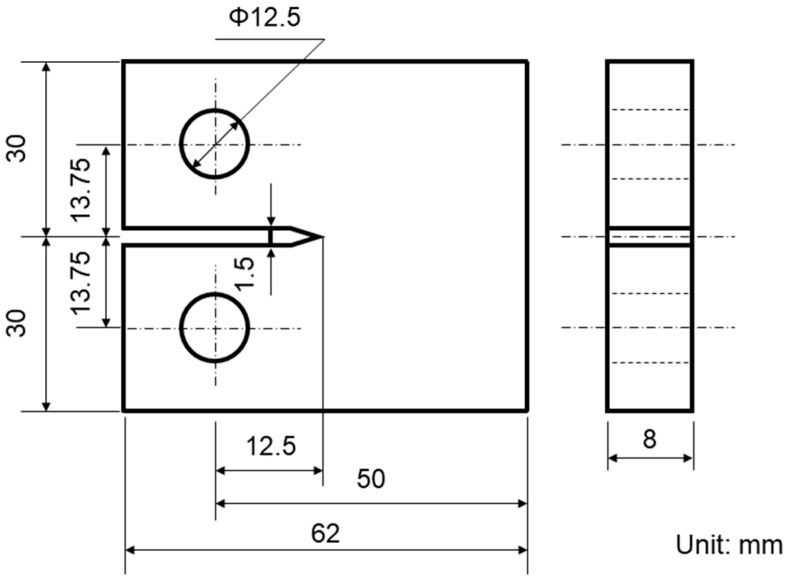
Sample dimensions for testing the fatigue crack propagation rate.

**Figure 5 materials-15-08610-f005:**
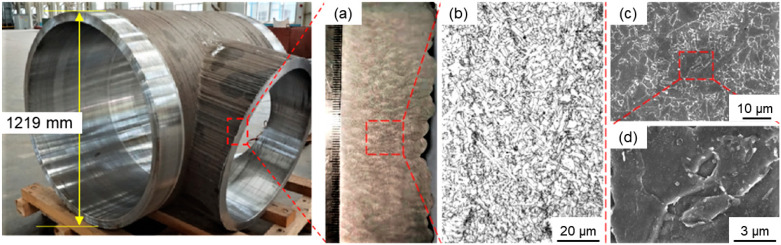
Microstructures in deposits of the SAAM workpieces. (**a**) Macro metallographic picture (**b**) Metallographic picture (**c**) Scanning electron microscope image (**d**) Local scanning electron microscope image.

**Figure 6 materials-15-08610-f006:**
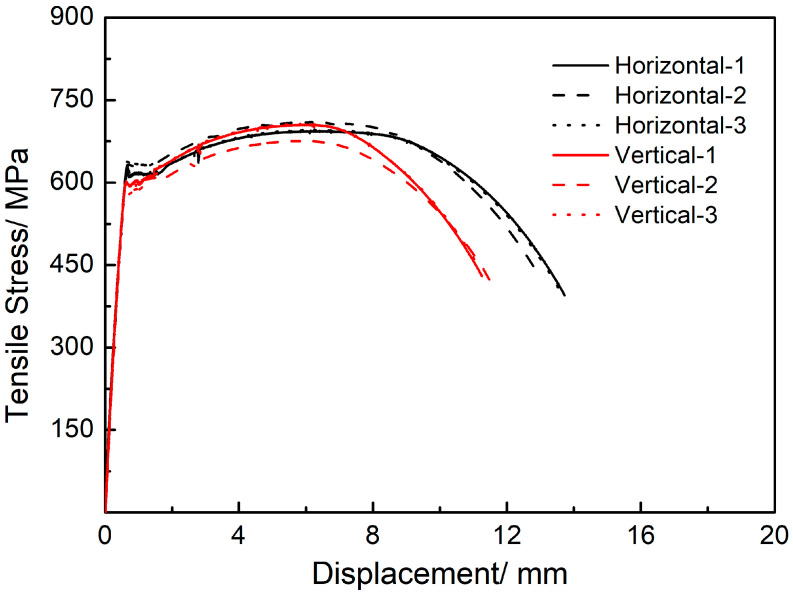
Tensile curves of SAAM workpieces along different directions.

**Figure 7 materials-15-08610-f007:**
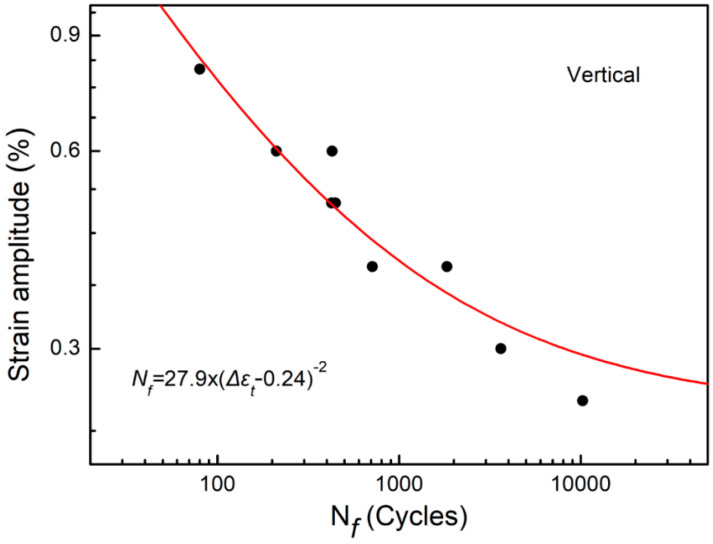
S-N curves of SAAM high-strength low-alloy steel workpieces along the vertical direction in low-cycle fatigue tests.

**Figure 8 materials-15-08610-f008:**
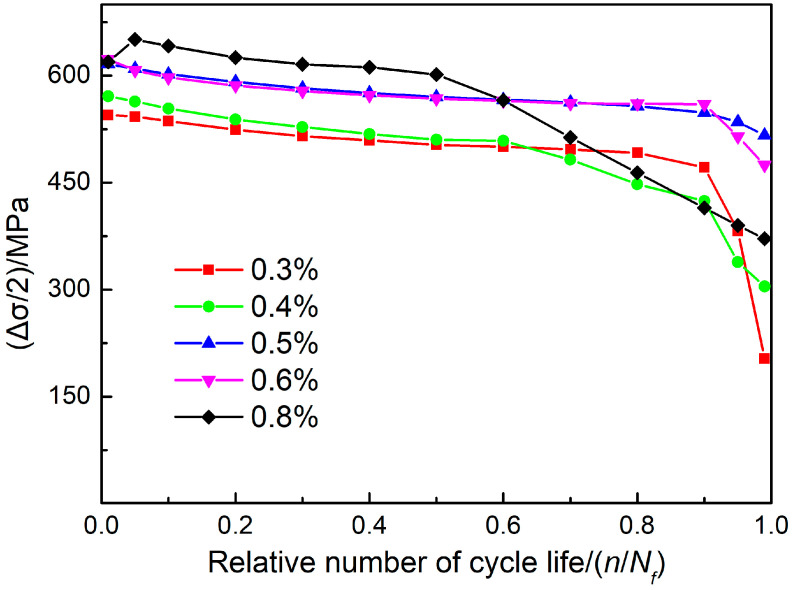
Changes in the amplitude of tensile stress of SAAM high-strength low-alloy steel workpieces under different strain amplitudes.

**Figure 9 materials-15-08610-f009:**
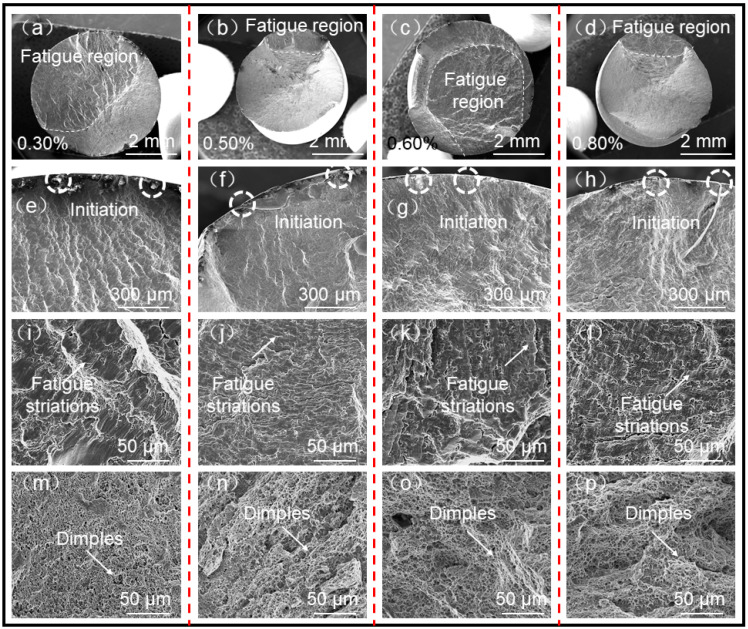
Low-cycle fatigue fractures of SAAM high-strength low-alloy steel workpieces under different strain amplitudes (**a**–**d**) are the macro fracture morphologies under different strain amplitudes respectively. (**e**–**h**) are fatigue crack initiation zones under different strain amplitudes respectively. (**i**–**l**) are the fatigue crack growth zones under different strain amplitudes respectively. (**m**–**p**) are the fracture zones under different strain amplitudes respectively).

**Figure 10 materials-15-08610-f010:**
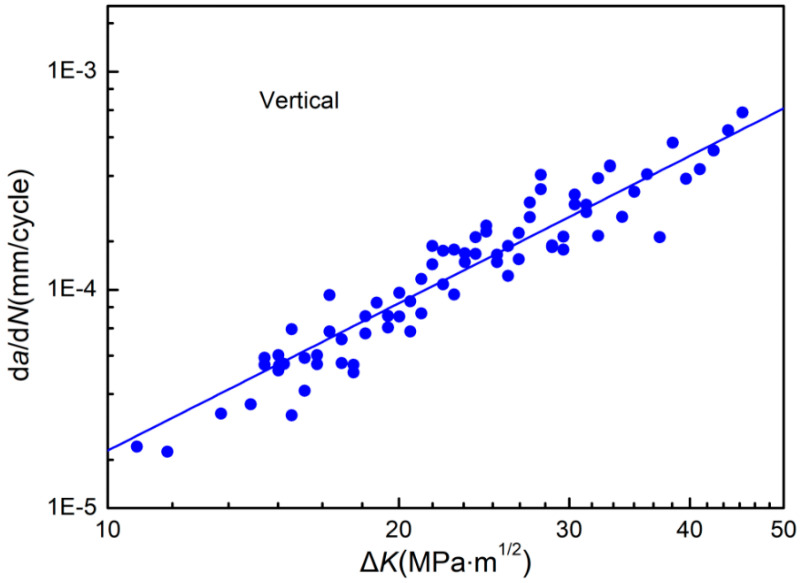
The relationship curve between the vertical fatigue crack propagation rate $\frac{da}{dN}$ of deposits of SAAM workpieces and the SIF amplitude ΔK.

**Figure 11 materials-15-08610-f011:**
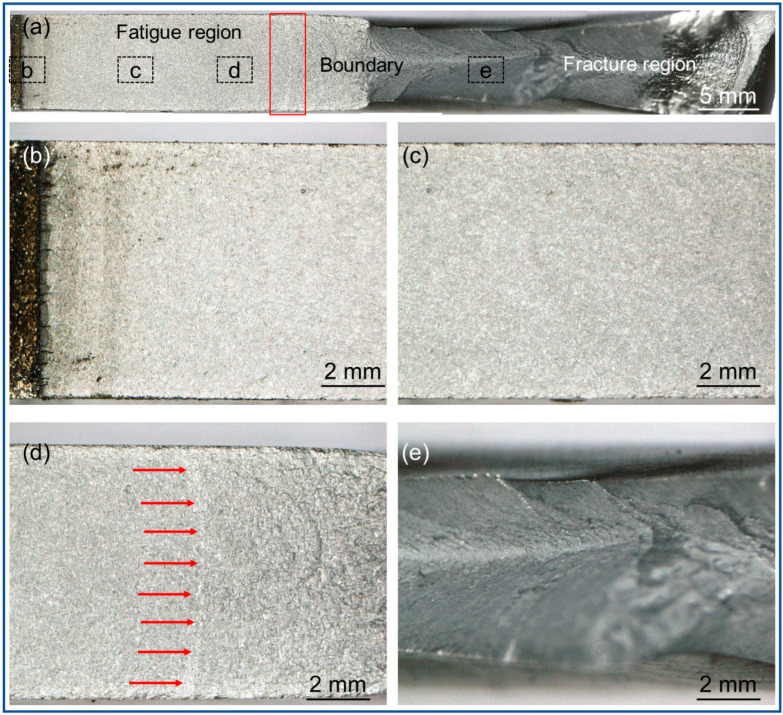
Fracture morphologies of samples for testing the fatigue crack propagation rate (**a**) is the macro fracture morphology; (**b**) is fatigue crack initiation zone; (**c**) is the fatigue crack stable growth zone; (**d**) is the fatigue crack instability propagation zone; (**e**) is the fault zone).

**Table 1 materials-15-08610-t001:** Chemical compositions of welding wires used for preparing high-strength low-alloy steel T-branch pipe through SAAM.

Elements	C	Si	Mn	P	S	Cr	Mo	Ni	Fe
Content	0.081	0.33	1.50	0.0052	0.0031	0.003	0.57	1.36	Balance

**Table 2 materials-15-08610-t002:** Additive manufacturing parameters for preparing high-strength low-alloy steel T-branch pipe via SAAM.

Voltage /V	Current /A	Velocity /(mm s^−1^)	Wire Feed Rate /(m min^−1^)	Interlayer Temperature /°C
28–35	500–700	6	1.35	140–200

**Table 3 materials-15-08610-t003:** Fatigue test data of the SAAM high-strength low-alloy steel workpieces under different strain amplitudes.

Strain Amplitude	0.8%	0.6%	0.5%	0.4%	0.3%	0.25%
/	80	428	425	712	3462	10,251
/	/	211	477	1837	/	/

## Data Availability

The data that support the findings of this study are available from the corresponding author, [author initials], upon reasonable request.

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
