# Peer review of "Microstructures and Fatigue Properties of High-Strength Low-Alloy Steel Prepared through Submerged-Arc Additive Manufacturing"

_materials, 2022, doi:10.3390/ma15238610_

Round 1

Reviewer 1 Report

Dear Authors,

The arc based additive manufacturing technology is very promising to fabricate high strength low-alloy steel due to the availability and ease of deposition. This paper “MICROSTRUCTURES AND FATIGUE PROPERTIES OF HIGH-STRENGTH LOW-ALLOY STEEL PREPARED THROUGH SUBMERGED-ARC ADDITIVE MANUFACTURING” investigates the microstructure and fatigue behaviour of HSLA grade via SAAM process. This paper contains a lot of technical details and is well-written. The following are the comments that should be addressed before acceptance. Minor revision is recommended.

Comment 1: In the introduction section, discussion related to WAAM should be improved and the following references may be used. Also, mention the benefits of using SAAM for fabricating HSLA components compared to other arc based WAAM processes.

1. Tiago A. Rodrigues, V. Duarte, Julian A. Avila, Telmo G. Santos, R.M. Miranda, J.P. Oliveira, Wire and arc additive manufacturing of HSLA steel: Effect of thermal cycles on microstructure and mechanical properties, Additive Manufacturing, Volume 27, 2019, Pages 440-450, https://doi.org/10.1016/j.addma.2019.03.029

2. Oleg Panchenko, Ivan Kladov, Dmitry Kurushkin, Leonid Zhabrev, Evgenii Ryl'kov, Maxim Zamozdra, Effect of thermal history on microstructure evolution and mechanical properties in wire arc additive manufacturing of HSLA steel functionally graded components, Materials Science and Engineering: A, Volume 851, 2022, 143569, https://doi.org/10.1016/j.msea.2022.143569

3. R. Sasikumar, A. Rajesh Kannan, S. Mohan Kumar, R. Pramod, N. Pravin Kumar, N. Siva Shanmugam, Yasam Palguna, Sakthivel Sivankalai, Wire arc additive manufacturing of functionally graded material with SS 316L and IN625: Microstructural and mechanical perspectives, CIRP Journal of Manufacturing Science and Technology, Volume 38, 2022, Pages 230-242, https://doi.org/10.1016/j.cirpj.2022.05.005

4. Jiarong Zhang, Chengning Li, Xiaocong Yang, Dongpo Wang, Wenbin Hu, Xinjie Di, Jianjun Zhang, In-situ heat treatment (IHT) wire arc additive manufacturing of Inconel625-HSLA steel functionally graded material, Materials Letters, Volume 330, 2023, 133326, https://doi.org/10.1016/j.matlet.2022.133326

Comment 2: The authors should mention the exact HSLA grade used in the present study and include the filler wire dimensions.

Comment 3: In Table 2, include the wire feed rate.

Comment 4: What does the following sentence mean? “The high-resolution scanning electron microscope (SEM) tests were to enhance corrosion on the basis of metallographic samples.” Correct the mistakes.

Comment 5: Why the stress ratio of -1 was considered for LCF tests? Also, mention the testing frequency. Any specific application? Why vertical direction was chosen for LCF specimen preparation?

Comment 6: Specify why a stress ratio of  0.1 was considered for CT specimen?

Comment 7: As shown in Fig.5, it looks like the final component. Does the author conducted any non-destructive tests and how were the results? If available include the same, it will be good for the readers.

Comment 8: During the LCF tests, mention the number of samples tested at each strain level?

Comment 9: Correct the mistake: table 1 should be 3 “Table 1 lists the fatigue test data of the SAAM workpieces under different strain amplitudes.”

Comment 10: Normally, during the fatigue tests with decreasing strain levels the crack propagation region should increase. Whether this trend was observed? Clarify.

Comment 11: The language of the manuscript should be polished throughout.

Author Response

Response to Reviewers

Response to Reviewer #1: (Highlighted with yellow color)

Comments:

In the introduction section, discussion related to WAAM should be improved and the following references may be used. Also, mention the benefits of using SAAM for fabricating HSLA components compared to other arc based WAAM processes.

  1. Tiago A. Rodrigues, V. Duarte, Julian A. Avila, Telmo G. Santos, R.M. Miranda, J.P. Oliveira, Wire and arc additive manufacturing of HSLA steel: Effect of thermal cycles on microstructure and mechanical properties, Additive Manufacturing, Volume 27, 2019, Pages 440-450, https://doi.org/10.1016/j.addma.2019.03.029
  2. Oleg Panchenko, Ivan Kladov, Dmitry Kurushkin, Leonid Zhabrev, Evgenii Ryl'kov, Maxim Zamozdra, Effect of thermal history on microstructure evolution and mechanical properties in wire arc additive manufacturing of HSLA steel functionally graded components, Materials Science and Engineering: A, Volume 851, 2022, 143569, https://doi.org/10.1016/j.msea.2022.143569
  3. R. Sasikumar, A. Rajesh Kannan, S. Mohan Kumar, R. Pramod, N. Pravin Kumar, N. Siva Shanmugam, Yasam Palguna, Sakthivel Sivankalai, Wire arc additive manufacturing of functionally graded material with SS 316L and IN625: Microstructural and mechanical perspectives, CIRP Journal of Manufacturing Science and Technology, Volume 38, 2022, Pages 230-242, https://doi.org/10.1016/j.cirpj.2022.05.005
  4. Jiarong Zhang, Chengning Li, Xiaocong Yang, Dongpo Wang, Wenbin Hu, Xinjie Di, Jianjun Zhang, In-situ heat treatment (IHT) wire arc additive manufacturing of Inconel625-HSLA steel functionally graded material, Materials Letters, Volume 330, 2023, 133326, https://doi.org/10.1016/j.matlet.2022.133326

Responses:

Agree.

Thanks for your comments.

These papers all mention that the traditional WAAM components are faced with a problem pertaining to inhomogeneity of microstructures and mechanical properties in various directions [1~4].

To solve the problem.

Due to the unique in-situ intrinsic heat treatment (IHT) typical of SAAM, that is, combining multi-layer penetration normalizing, full-layer-penetration inter-critical annealing, and long-duration tempering, the middle part of deposited walls is mainly composed of completely equiaxed ferrites that are homogeneous in the vertical direction.

Compared with WAAM components, the mechanical properties of SAAM components exhibit lower anisotropy and better strength-toughness balance.

  1. Tiago A. Rodrigues, V. Duarte, Julian A. Avila, Telmo G. Santos, R.M. Miranda, J.P. Oliveira, Wire and arc additive manufacturing of HSLA steel: Effect of thermal cycles on microstructure and mechanical properties, Additive Manufacturing. 2019;27:440-450.
  2. Oleg P, Ivan K, Dmitry K, Leonid Z, Evgenii R, Maxim Z, Effect of thermal history on microstructure evolution and mechanical properties in wire arc additive manufacturing of HSLA steel functionally graded components, Materials Science and Engineering: A. 2022;851:143569.
  3. R. Sasikumar, A. Rajesh Kannan, S. Mohan Kumar, R. Pramod, N. Pravin Kumar, N. Siva Shanmugam, Yasam Palguna, Sakthivel Sivankalai, Wire arc additive manufacturing of functionally graded material with SS 316L and IN625: Microstructural and mechanical perspectives, CIRP Journal of Manufacturing Science and Technology. 2022;38:230-242.
  4. Jiarong Zhang, Chengning Li, Xiaocong Yang, Dongpo Wang, Wenbin Hu, Xinjie Di, Jianjun Zhang, In-situ heat treatment (IHT) wire arc additive manufacturing of Inconel625-HSLA steel functionally graded material, Materials Letters. 2023;330:133326

Please see:

Lines 52-56, 58~62, 65~67 and 312~324.

2.

Comments:

The authors should mention the exact HSLA grade used in the present study and include the filler wire dimensions.
Responses:

Agree.

Thanks for your comments.

The HSLA grade is 550MPa and the diameter of the welding wire is 4 mm.

We added relevant information.

Please see:

Line 99.

3.

Comments:

In Table 2, include the wire feed rate.

Responses:

Agree.

Thanks for your comments.

The wire feed rate is 1.35m/min.

Please see:

Line 102.

4.

Comments:

What does the following sentence mean? “The high-resolution scanning electron microscope (SEM) tests were to enhance corrosion on the basis of metallographic samples.” Correct the mistakes.

Responses:

Agree.

Thanks for your comments.

We have revised this sentence.

Please see:

Lines 106~107.

5.

Comments:

Why the stress ratio of -1 was considered for LCF tests? Also, mention the testing frequency. Any specific application? Why vertical direction was chosen for LCF specimen preparation?

Responses:

Agree.

Thanks for your comments.

When the pressure inside the pipe is low, the inner wall of the pipe is subjected to compressive stress; When the pressure inside the pipe is large, the inner wall of the pipe is subjected to tensile stress. So, the strain ratio was chosen as -1.

The frequency in the experiment was 0.5Hz per second.

According to the experimental results, the tensile strength of the vertical is close to that of the horizontal, but the plasticity is worse than that of the horizontal. Therefore, the vertical sample is selected for low cycle fatigue test.

Please see:

Line 133.

6.

Comments:

Specify why a stress ratio of 0.1 was considered for CT specimen?

Responses:

Agree.

Thanks for your comments.

In the experiment of fatigue crack growth rate, tensile stress makes a great contribution to fatigue crack growth, while compressive stress makes almost no contribution, so the stress ratio of 0.1 is selected.

7.

Comments:

As shown in Fig.5, it looks like the final component. Does the author conducted any non-destructive tests and how were the results? If available include the same, it will be good for the readers.

Responses:

Agree.

Thanks for your comments.

We conducted non-destructive testing of the manufactured components and found no defects.

8.

Comments:

During the LCF tests, mention the number of samples tested at each strain level?

Responses:

Agree.

Thanks for your comments.

In this experiment, when the strain amplitude were 0.6%, 0.5% and 0.4%, two samples were tested. When the strain amplitudes were 0.8%, 0.3% and 0.25%, a sample was tested.

We added this information to the article.

Please see:

Lines 127~129.

9.

Comments:

Correct the mistake: table 1 should be 3 “Table 1 lists the fatigue test data of the SAAM workpieces under different strain amplitudes.”

Responses:

Agree.

Thanks for your comments.

According to your suggestion, we have made a correction.

Please see:

Lines 182~183.

10.

Comments:

Normally, during the fatigue tests with decreasing strain levels the crack propagation region should increase. Whether this trend was observed? Clarify.

Responses:

Agree.

Thanks for your professional comments.

This article does not see this trend.

Your comments are very professional. Theoretically, as the strain level decreases, the crack growth zone in the fatigue fracture will increase. However, in the process of fatigue experiment, in order to prevent the extensometer and equipment used to measure real-time strain from being damaged, the experiment will be stopped when the load drops by 25% according to the experimental standard. At this time, the sample has fatigue cracks but is not broken, so as to better avoid the oxidation of fatigue fracture. After the fatigue experiment is completed, the fatigue sample is finally pulled off. At the same time, the low-cycle fatigue experiment in this paper mainly focuses on the initiation life of fatigue microcracks, while the fatigue crack growth related problems are the focus of the fatigue crack growth rate experiment in the following paper.

Your opinion is very good, we will focus on this problem in the future research.

11.

Comments:

The language of the manuscript should be polished throughout.

Responses:

Agree.

Thank you for your suggestion. We have checked the full text and changed many long sentences into short sentences.

Please see:

Lines 9~10, 50~51, 88~89, 133, 143~144, 215~216.

Reviewer 2 Report

The authors studied the microstructures and fatigue properties of high-strength low-alloy steel prepared through submerged-arc additive manufacturing. The manuscript had an interesting topic and was well-written, however it could only be accepted with the following minor revisions:

1.    Introduction, problem statement, and summary of the study were not found in the abstract.

2.    The study focuses on SAAM technology and additive manufacturing (AM) in general. It's wonderful that AM could be introduced, with its advantages and limitations. You may refer to the related paper on AM such as Rheological Properties of Natural Fiber Reinforced Thermoplastic Composite for Fused Deposition Modeling (FDM): A Short Review, Journal of Advanced Research in Fluid Mechanics and Thermal Sciences, 98(2), 157-164.

3. SAAM’s machine, specification, brand, model, manufacturer, and samples were not shown in the study. Please include those samples and figures of the experimental setup.

4.    For section 2, it is recommended that the authors provide a flow chart for the experimental setup and materials.

5. “CT sample dimensions, initial crack length, and length of initially prefabricated fatigue crack were 62 mm × 60 mm × 8 mm, 12.5 mm, and 2.5 mm, respectively”. Those dimensions were unnecessary as already stated in Figure 4.

6.    Labeling in Figure 4(a), (b) and (c) were very small (please enlarge).

7. It is suggested to add the limitation, future study and develop implications for researchers in the conclusion section.

Author Response

Response to Reviewers

Response to Reviewer #2: (Highlighted with red color)

Comments:

Introduction, problem statement, and summary of the study were not found in the abstract.

Responses:

Agree.

Thanks for your comments.

According to your suggestion, we have revised the abstract.

Please see:

Lines 10~11, 20~23.

Comments:

The study focuses on SAAM technology and additive manufacturing (AM) in general. It's wonderful that AM could be introduced, with its advantages and limitations. You may refer to the related paper on AM such as Rheological Properties of Natural Fiber Reinforced Thermoplastic Composite for Fused Deposition Modeling (FDM): A Short Review, Journal of Advanced Research in Fluid Mechanics and Thermal Sciences, 98(2), 157-164.

Responses:

Agree.

Thanks for your comments.

Mohd NA et al. made a very systematic summary of the advantages and disadvantages of additive manufacturing technology.

  1. Mohd NA, Mohamad RI, Mastura MT, Faizal M, Zulkiflle L. Rheological Properties of Natural Fiber Reinforced Thermoplastic Composite for Fused Deposition Modeling (FDM): A Short Review, Journal of Advanced Research in Fluid Mechanics and Thermal Sciences. 2022;98(2):157-164.

Please see:

Lines 39~41, 304~306.

Comments:

SAAM’s machine, specification, brand, model, manufacturer, and samples were not shown in the study. Please include those samples and figures of the experimental setup.

Responses:

Agree.

Thanks for your comments.

Figure 5 shows pictures of submerged arc welding additive manufacturing samples.

I'm sorry that our submerged arc welding additive manufacturing equipment is independently developed and designed.

Please see:

Line 164.

Comments:

For section 2, it is recommended that the authors provide a flow chart for the experimental setup and materials.

Responses:

Thanks for your comments.

I'm sorry that our submerged arc welding additive manufacturing equipment is independently developed and designed.

Comments:

“CT sample dimensions, initial crack length, and length of initially prefabricated fatigue crack were 62 mm × 60 mm × 8 mm, 12.5 mm, and 2.5 mm, respectively”. Those dimensions were unnecessary as already stated in Figure 4.

Responses:

Agree.

Thanks for your comments.

According to your suggestion, we have revised the manuscript.

Please see:

Lines 143~144.

Comments:

Labeling in Figure 4(a), (b) and (c) were very small (please enlarge).

Responses:

Agree.

Thanks for your comments.

According to your suggestion, we have revised the Figure 4(a), (b) and (c).

Please see:

Line 164.

Comments:

It is suggested to add the limitation, future study and develop implications for researchers in the conclusion section.

Responses:

Agree.

Thanks for your comments.

According to your suggestion, we have added this part.

How to control the interlayer temperature of multi-layer multi-channel submerged arc welding additive and obtain a relatively uniform microstructure in all directions is one of the key factors to improve the fatigue performance of additive manufacturing parts.

Please see:

Lines 261~263.

Reviewer 3 Report

Please try to use shorten sentences as many are very long and difficult to read and take their message. AN example is first sentence in abstract. Other were found the same in many parts of the manuscript

was very approximate”- what that means?

“while the plasticity in vertical direction” this is a wrong observation !

The abstract should be reformulated as no much sense were noted in

Not clear which is the point of discussion X70 and X 80 against “prepared 625 nickel-base alloy components”. IT seems you mix the materials and not a proper flow of introduction were found.

Which is actually challenge that the research community face in order to introduce your research ?

I strongly advice to revise the introduction as not make much sense !

The authors contribution should be well highlighted otherwise difficult to see the novelty of this work !

In Figure 1 you showed well different type of samples used but actually why you use different parts for different type of characterization ? Cause it is assumed you do a characterization of mechanical test and then on that sample you make the characterization of microstructural behaviour !

Also in material you mix the tensile with fatigue test

Therefore I advice to revise this work and then I will be able to review correctly this work

The initiation of fracture in Figure 11 b is not correct actually !

Author Response

Response to Reviewers

Response to Reviewer #3: (Highlighted with pink color)

Comments:

Please try to use shorten sentences as many are very long and difficult to read and take their message. An example is first sentence in abstract. Other were found the same in many parts of the manuscript.

Responses:

Agree.

Thanks for your comments.

We check the language of the manuscript and revise it, changing long sentences to short sentences. Thank you for your suggestion, we will try our best to use short sentences in future manuscript writing.

Please see:

Lines 9~10, 50~51, 133, 143~144, 215~216.

Comments:

“was very approximate”- what that means?

Responses:

Agree.

Thanks for your comments.

Results show that the horizontal tensile strength of deposits was quite close to the vertical one.

There was a mistake in the expression, and we revised it.

Please see:

Lines 14~15.

3

Comments:

“while the plasticity in vertical direction” this is a wrong observation!

Responses:

Agree.

Thanks for your comments.

while the elongation rate in vertical direction was obviously lower than that in horizontal direction.

We have revised this sentence.

Please see:

Line 15.

4

Comments:

The abstract should be reformulated as no much sense were noted in

Responses:

Agree.

Thanks for your comments.

According to your suggestion, we have revised the abstract.

Please see:

Lines 9~10, 20~23.

5

Comments:

Not clear which is the point of discussion X70 and X 80 against “prepared 625 nickel-base alloy components”. IT seems you mix the materials and not a proper flow of introduction were found.

Responses:

Agree.

Thanks for your comments.

The purpose of our research results on these materials is to illustrate that for wire and arc additive manufacturing (WAAM), it is very easy to have inhomogeneity in the microstructure of the additive.

Please see:

Lines 50~52.

6

Comments:

Which is actually challenge that the research community face in order to introduce your research ?

Responses:

Agree.

Thanks for your comments.

Traditional WAAM components are faced with a problem pertaining to inhomogeneity of microstructures and mechanical properties in various directions.

To solve the problem.

Due to the unique in-situ intrinsic heat treatment (IHT) typical of SAAM, that is, combining multi-layer penetration normalizing, full-layer-penetration inter-critical annealing, and long-duration tempering, the middle part of deposited walls is mainly composed of completely equiaxed ferrites that are homogeneous in the vertical direction.

Compared with WAAM components, the mechanical properties of SAAM components exhibit lower anisotropy and better strength-toughness balance.

Please see:

Lines 52-56, 58~62, 65~67.

7

Comments:

I strongly advice to revise the introduction as not make much sense !

Responses:

Agree.

Thank you very much for your advice, you have very high requirements for the way the article is written. Your high standards give us the motivation to progress in our research. We have rearranged the logic of the introduction and removed some redundant content.

Please see:

Lines 50-52, 88~89.

8

Comments:

The authors contribution should be well highlighted otherwise difficult to see the novelty of this work!

Responses:

Agree.

Thanks for your comments.

At present, manufacturing high-strength low-alloy steel T-branch pipe through SAAM has not been reported.

Please see:

Lines 88-89.

9

Comments:

In Figure 1 you showed well different type of samples used but actually why you use different parts for different type of characterization? Cause it is assumed you do a characterization of mechanical test and then on that sample you make the characterization of microstructural behavior!

Responses:

Agree.

Thanks for your comments.

Your suggestions are very professional and you have presented a very reasonable logical structure of the paper. However, due to the COVID-19 pandemic, there is no way to supplement the microscopic experiments of different parts. At the same time, this paper focuses on the mechanical properties of additive parts. Your suggestions have inspired us, and we will focus on adopting them in the following research. Thank you very much!

10

Comments:

Also in material you mix the tensile with fatigue test.

Responses:

Thanks for your comments.

Tensile test is to study the basic properties of the material, while fatigue test is to study the service performance of the material.

In the logical structure of this paper, the tensile experiment is to lay the groundwork for the subsequent fatigue experiment. In this paper, the vertical specimens with weak tensile properties were selected by tensile test first, and then the fatigue test of vertical specimens were carried out.

11

Comments:

Therefore I advice to revise this work and then I will be able to review correctly this work. The initiation of fracture in Figure 11 b is not correct actually!

Responses:

Agree.

Thanks for your comments.

Thank you very much for taking the time to review our paper.

We have revised Figure 11(b).

Please see:

Line 244.

Round 2

Reviewer 2 Report

The authors have done the correction according to the comments. 

Reviewer 3 Report

.